# Design and Optimization of Lightweight Lithium-Ion Battery Protector with 3D Auxetic Meta Structures

**Michael Alfred Stephenson Biharta [1]**, **Sigit Puji Santosa [2,3,\*]**, **Djarot Widagdo [2]** and **Leonardo Gunawan [2,3]**

1 Department of Aerospace Engineering, Faculty of Mechanical and Aerospace Engineering, Institut Teknologi Bandung (ITB), Jalan Ganesha 10, Bandung 40132, Indonesia; michael.asb@students.itb.ac.id
2 Lightweight Structures Research Group, Faculty of Mechanical and Aerospace Engineering, Institut Teknologi Bandung (ITB), Jalan Ganesha 10, Bandung 40132, Indonesia; dwidagdo@ae.itb.ac.id (D.W.); gun@ftmd.itb.ac.id (L.G.)
3 National Center for Sustainable Transportation Technology (NCSTT), Institut Teknologi Bandung (ITB), Jalan Ganesha 10, Bandung 40132, Indonesia
\* Correspondence: sigit.santosa@itb.ac.id

**Abstract:** This research study involves designing and optimizing a sandwich structure based on an auxetic structure to protect the pouch battery system for electric vehicles undergoing ground impact load. The core of the sandwich structure is filled with the auxetic structure that has gone through optimization to maximize the specific energy absorbed. Its performance is analyzed with the non-linear finite element method. Five geometrical variables of the auxetic structures are analyzed using the analysis of variance and optimized using Taguchi's method. The optimum control variables are double-U hierarchal (DUH), the cross-section's thickness = 2 mm, the length of the cell = 10 mm, the width of the cell = 17 mm, and the bending height = 3 mm. The optimized geometries are then arranged into three different sandwich structure configurations. The core is filled with optimized DUH cells that have been enlarged to 200% in length, arranged in $11 \times 11 \times 1$ cells, resulting in a total dimension and mass of $189 \times 189 \times 12$ mm and 0.75 Kg. The optimized sandwich structure shows that the pouch battery cells can be protected very well from ground impact load with a maximum deformation of 1.92 mm, below the deformation threshold for battery failure.

**Keywords:** auxetic structure; battery protection; crashworthiness; Taguchi's method





## 1. Introduction

Lithium-ion batteries currently play an essential part in the modern world. This type of battery is commonly used as an energy component of consumer electronic equipment. It continues to experience rapid development in various aspects, such as energy density, weight, and manufacturing methods [1]. Recently, lithium-ion batteries have also begun to be used as energy storage in transportation fields, such as electric cars and planes, in the transportation industry's efforts to reduce $CO_2$ emissions in the environment [2]. According to Novizayanti et al. [3], the three most prioritized factors in choosing electric cars in Indonesia are vehicle range, price, and speed. This new field of use and demand from the market poses new safety challenges due to the difference in external elements that the lithium-ion battery will experience during operation and can potentially cause accidents and danger to passengers. A traffic accident could expose both passengers and the rescue party to new hazards [4]. Burning is the most common accident associated with batteries in both cars and planes [5,6].

Many of the cases are post-crash fires. Post-crash fires are caused by the battery's thermal runaway, a short circuit between the battery's different components [7]. The short circuit is caused by physical contact between the battery components caused by battery deformation due to the crash [8]. Fire in lithium-ion batteries can pose safety risks to emergency responders because it can reignite even after an initial fire is put out [9]. With

the growing interest in lithium-ion batteries as energy storage, there is an urgent need to ensure safety to reduce post-crash fire risk by finding the optimum battery protector.

The battery protector must have high energy absorption capability and high strength to ensure that the protected battery does not experience excessive deformation. It is also preferable to be as light as possible because a lighter vehicle consumes less energy under the same condition [10]. Various kinds of structures have been investigated as lithium-ion battery protectors, such as sandwich structures. These structures are lightweight, have high energy absorption capabilities, and are quite common in aerospace and construction [11,12].

The core of the sandwich structure can be filled with various structures, such as a meta structure [12]. Meta structure is an arrangement of unit cell structures arranged repeatedly, resulting in a lighter structure than a comparable solid structure. Examples in nature are human bones and beehive structures, usually called honeycombs. Meta structure can be further divided into many structures, such as auxetic, lattice, and chiral [13].

An auxetic structure is a type of meta structure where each cell structure has a negative Poisson's ratio. This unique property means that this structure has several advantages, such as the ability to generate a curved surface mainly distributed by positive Gaussian curvature when subjected to out-of-plane bending (which is the direction of the load to the battery), improved resistance against shear deformation, improved indentation resistance at points where concentrated loads are applied, and a high quantity of energy absorption and damping, all while keeping its weight down [13–15].

This research will study battery protectors with sandwich-based 3D auxetic structures with in-plane impact, specifically ground impact (based on reference Xia [16]). Four 3D auxetic shapes will be analyzed: double-arrowed [17], double-U [18], re-entrant A [19–21], and re-entrant B [19–21]. All numerical results in this study were derived using the non-linear dynamics with the finite element method [22].

Optimization of the lattice structure for maximum specific energy absorption has been previously studied by Nasrullah et al. [23], where an optimum twisted lattice structure was found to have the best results in specific energy absorption. Optimization of the sandwich-based structure with a composite core for battery protection has been previously studied by Irawan et al. [24], where the design provided good safety for the electric vehicle's battery. Optimization of the sandwich-based lattice structure for battery protection has been previously studied by Pratama et al. [25] and M. Z. Mahasin [26], where an optimum lattice structure could be applied as the sandwich core in the battery protection system and minimize battery deformation. This study will vary by offering more choices for lithium-ion battery protection, specifically pouch battery (based on reference Sahraei et al. [27]). Moreover, the data from this study may be used for another crashworthiness application using the 3D auxetic structure.

## 2. Cell Materials, Methods, and Result

### 2.1. Cell Validation

2.1.1. Cell Validation Modeling

Double-U is one of the candidate auxetic structures used as the core of the sandwich structure to protect the battery. This shape is chosen to validate the numerical modeling of all auxetic structures in this research. The double-U structure's numerical model and experimental result are based on reference Yang [18].

The general configuration of the validating simulation consists of (from top to bottom) the impactor, double-U cell, and floor, where the distance between each part is set to be as close as possible without pre-penetration. A rigid shell plane with element size of 1 mm and thickness of 1 mm is used as the impactor and floor. Figure 1 illustrates the general configuration of this model.

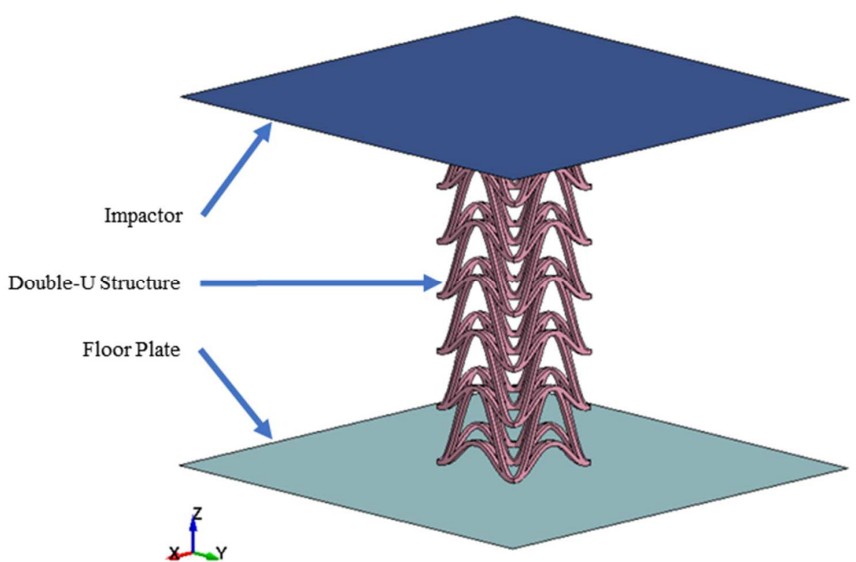

**Figure 1.** The general configuration of double-U structure simulation; impactor as blue plane, double-U structure as pink solid, floor as turquoise plane.

Here, the whole structure consists of 6 individual cells stacked on top of each other ($1 \times 1 \times 6$ unit cells). Table 1 summarizes the sizes of the structural cell, and Figure 2 explains the meaning of each dimension parameter. The structure was modeled using fully integrated solid elements with piecewise linear plasticity material. It was meshed using multi-solids meshing tools from Hypermesh with element size of 0.33 mm (thickness of the cell divided by 3) and source shells mixed.

**Table 1.** Double-U cell's geometry [18].

| Parameter | Value | Unit |
|:---:|:---:|:---:|
| $\theta_1$ | 61 | degree |
| $\theta_2$ | 31 | degree |
| $h_1$ | 18 | mm |
| $h_2$ | 6 | mm |
| $l$ | 10 | mm |
| $b$ | 1 | mm |
| Thickness | 1 | mm |

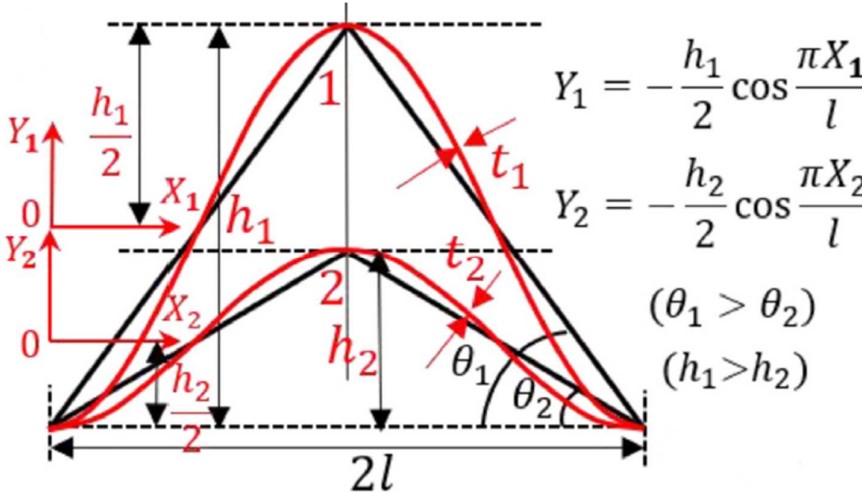

**Figure 2.** Geometry parameters of the 2D double-U structure [18].

The simulation is performed in a quasistatic process, so the impactor's load is applied as prescribed motion velocity. The velocity will increase according to Table 2, with the final velocity of 1 m/s in the direction of $Z - (\textbf{DOF} = 3)$. The velocity data are applied to the impactor as prescribed motion. The movement of every node of floor elements is constrained in every direction, and the movement of every node of impactor elements is constrained in every direction but the Z direction.

**Table 2.** Velocity data.

| Time (ms) | Velocity (According to Final Velocity) |
|---|---|
| 0 | 0 |
| 1 | 10% |
| 2 | 90% |
| 3 | 100% |

In the experiment of reference Yang [18], the cell is fabricated through 3D printing using selective laser melting technology with stainless steel SS316L as the material. Table 3 and Figure 3 will summarize the material property of stainless steel SS316L.

**Table 3.** SS316L material properties [18].

| Variable | Value | Unit |
|---|---|---|
| Density $(\rho)$ | $8 \times 10^{-6}$ | $Kg/mm^3$ |
| Modulus of Elasticity $(E)$ | 200 | GPa |
| Yield Strength $(\sigma_{ys})$ | 500 | MPa |
| Poisson's Ratio $(v)$ | 0.3 | |

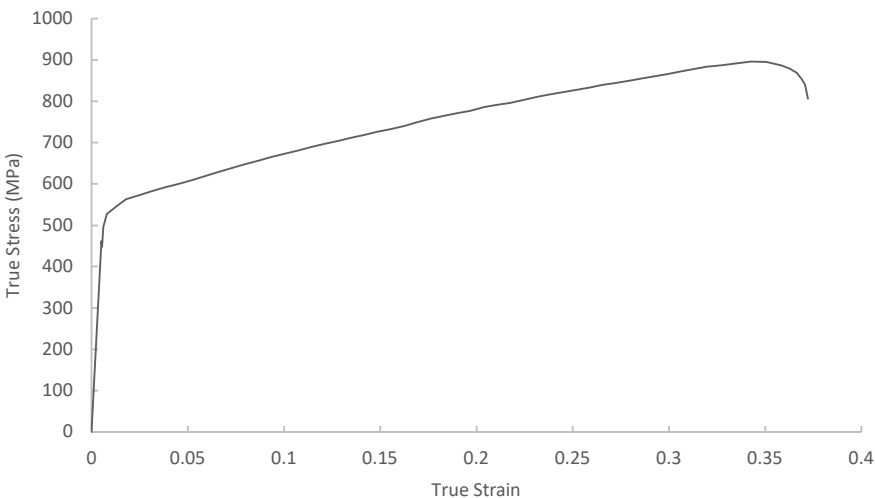

**Figure 3.** Experimental true stress–true strain curve of SS316L [18].

Automatic single surface was applied for the contact between elements from the same part (cell's structure). Automatic surface to surface was applied for the contact between different parts (cell–impactor and cell–floor).

### 2.1.2. Cell Validation Result

There is a significant difference between the numerical model and the experiment condition. The experiment object is configured in $4 \times 4 \times 6$ unit cells (compared to $1 \times 1 \times 6$ unit cells in the numerical model). This change was performed because of the limitations of computing power that the writer had access to. To combat this difference, the results from both the experimental and numerical models are divided by their footprint

area (area of the floor that the specimen occupies), resulting in a pressure–displacement curve displayed in Figure 4.

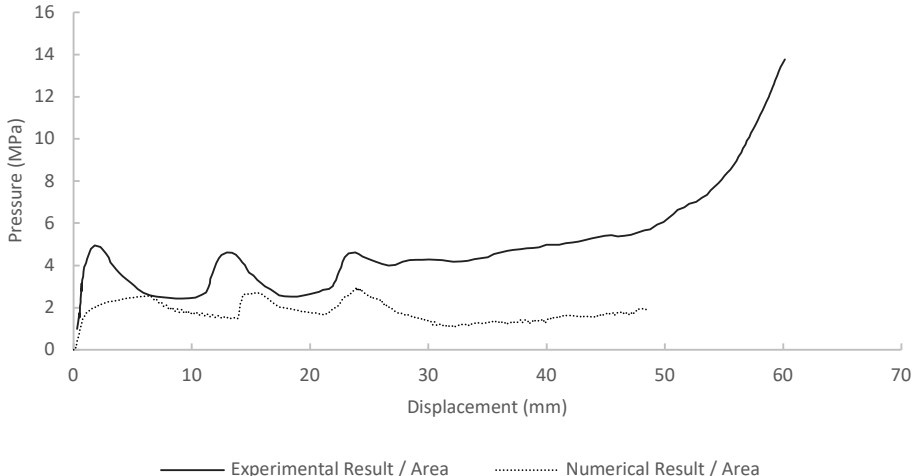

**Figure 4.** Pressure curve of double-U structure.

At the first look, it is seen that the result is not similar; the pressure of the experimental result is twice the numerical result. However, the difference in pressure between the reference result and the writer's result can be attributed to the side-to-side interaction in the reference model that prevents buckling, which is nonexistent in the writer's numerical model. Because the position of each peak is very close, the writer's result can be considered accurate enough for this research. Figure 5 shows the difference between the references and the writer's result.

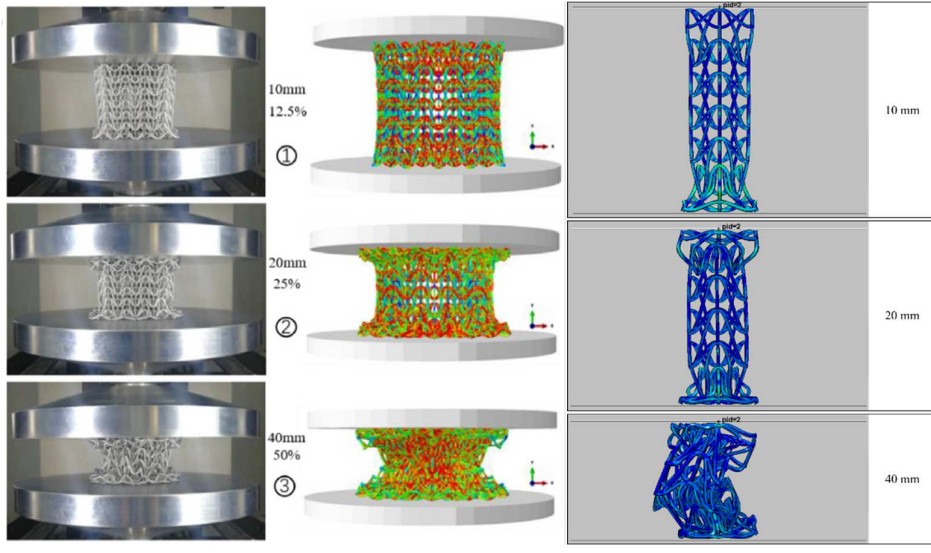

**Figure 5.** Reference's experimental results (**left**), reference's numerical results (**middle**), and writer's numerical results (**right**) of quasistatic compression of double-U structure.

*2.2. Cell Modeling*

2.2.1. Geometry and Meshing

The modeling of single-cell structure optimization is based on Section 2.1's model with some changes. The optimization model's general configuration and boundary condition stay the same as the validation model. Firstly, this simulation only analyzes single-cell structures from each geometric shape: double-arrowed [17], double-U [18], re-entrant A [19–21], and re-entrant B [19–21]. Because of the difference in geometric parameters

between each geometric shape, there is a need to create new geometric parameters that can accommodate all of the geometric shapes in this research to ease the optimization process. It was decided that four parameters are enough to explain the size of all geometric shapes, which are length (L), width (W), bending height (H), and thickness of the cross-section. The cross-section of all cells in this research is limited to an equal-sided square. Figure 6 illustrates the new geometric parameters.

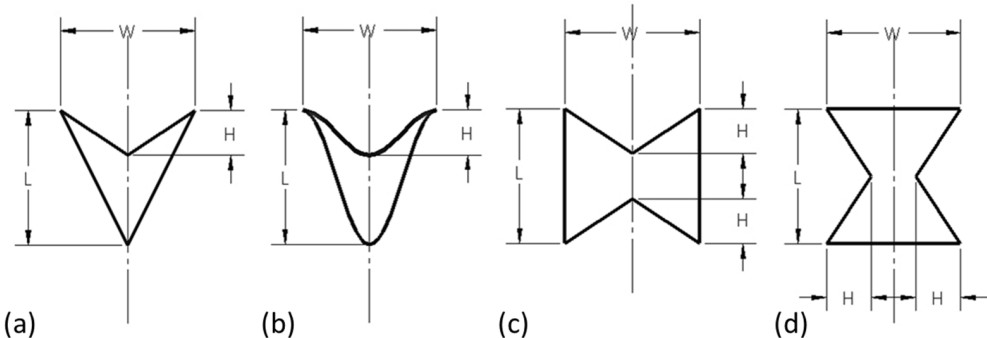

**Figure 6.** New geometric parameters, DAA (**a**), DUH (**b**), RE-A (**c**), RE-B (**d**).

### 2.2.2. Orthogonal Array

The optimization aims to produce a geometry with the highest specific energy absorbed (SEA) during the compressive load. The Taguchi orthogonal array determines the variation of the cell's geometry in each run. Five variables are chosen as the control factor, which are geometry, length (L), width (W), bending height (H), and thickness, with four levels for each variable. Table 4 will summarize the variables and levels of the parameters.

**Table 4.** Auxetic structure parameters.

| Index | Variables | Level | | | |
|---|---|---|---|---|---|
| | | **1** | **2** | **3** | **4** |
| A | Geometry | DAA | DUH | RE-A | RE-B |
| B | L (mm) | 9 | 10 | 11 | 12 |
| C | W (mm) | 15 | 16 | 17 | 18 |
| D | H (mm) | 1.5 | 2 | 2.5 | 3 |
| E | Thickness (mm) | 1 | 1.33 | 1.66 | 2 |

Based on the number of variables and levels, the smallest possible Taguchi orthogonal array is L16′, which means there are 16 different models. The Taguchi orthogonal design for this research is based on reference Pratama et al. [25] with three noise levels. The noise variables are limited to the impactor's velocity and the material's density, both with an error of ±5% from nominal, as shown in Table 5. Noises are designed to simulate real-world disturbances that may affect the result of the experiment.

**Table 5.** Noise variables and levels.

| Variables | Noise | | |
|---|---|---|---|
| | **Increasing (+1)** | **Nominal** | **Reducing (−1)** |
| Impactor Velocity (m/s) | 5.25 | 5 | 4.75 |
| Material Density | 105% Rho | 100% Rho | 95% Rho |

This results in 48 numerical analyses, consisting of 16 different models, and each model is analyzed three times for each level of noise, summarized in Table 6.

**Table 6.** Taguchi orthogonal design of the experiment.

| Model No. | Factors | | | | |
|---|---|---|---|---|---|
| | Geometry | L (mm) | W (mm) | H (mm) | Thickness (mm) |
| 1 | DAA | 9 | 15 | 1.5 | 1 |
| 2 | DAA | 10 | 16 | 2 | 1.33 |
| 3 | DAA | 11 | 17 | 2.5 | 1.67 |
| 4 | DAA | 12 | 18 | 3 | 2 |
| 5 | DUH | 9 | 18 | 2 | 1.67 |
| 6 | DUH | 10 | 17 | 1.5 | 2 |
| 7 | DUH | 11 | 16 | 3 | 1 |
| 8 | DUH | 12 | 15 | 2.5 | 1.33 |
| 9 | RE-A | 9 | 17 | 3 | 1.33 |
| 10 | RE-A | 10 | 18 | 2.5 | 1 |
| 11 | RE-A | 11 | 15 | 2 | 2 |
| 12 | RE-A | 12 | 16 | 1.5 | 1.67 |
| 13 | RE-B | 9 | 16 | 2.5 | 2 |
| 14 | RE-B | 10 | 15 | 3 | 1.67 |
| 15 | RE-B | 11 | 18 | 1.5 | 1.33 |
| 16 | RE-B | 12 | 17 | 2 | 1 |

2.2.3. Material and Contact Modeling

The material model is added with a failure condition to enhance the realistic foundation for this research further. All the cell's material is changed into titanium alloy Ti-6AL-4V with material properties as shown in Table 7.

**Table 7.** Ti-6Al-4V material properties.

| Variable | Value | Unit |
|---|---|---|
| Density ($\rho$) [28] | $4.43 \times 10^{-6}$ | Kg/mm$^3$ |
| Modulus of Elasticity ($E$) [28] | 110.32 | GPa |
| Yield Strength ($\sigma_{ys}$) [29] | 0.93 | GPa |
| Poisson's Ratio ($v$) [28] | 0.31 | |
| Failure Strain ($\varepsilon_f$) [30] | 0.2 | |
| Cowper Symond's Constant (D) [30] | 200 | |
| Cowper Symond's Constant (q) [30] | 15 | |

The failure strain is inputted into the failure material model, which provides a way of including failure into the existing material model as the maximum effective strain at failure. Because of the added failure material model, the automatic single surface contact is replaced by the eroded single surface contact, and the automatic surface to surface contact is replaced by the eroded surface to surface contact, both with pinball segment-based contact and warped segment checking + sliding option.

Time termination is calculated for every run where the maximum displacement is equal to 70% of the cell's length (parameter length, not physical length, which changes according to thickness), which follows Equation (1).

$$T_{termination} = \frac{70\% \, L}{V_{final}} + 0.15 \tag{1}$$

*2.3. Cell Optimization Result and Discussion*

Taguchi's optimization and analysis of variance were used to find the most optimum geometry and the contribution of each variable to the SEA of the structure based on the result of 48 numerical analyses (see Section 2.2.2). It is found that the most optimum geometric parameters for a cell model to obtain the highest SEA during a compression load are summarized in Figure 7 and Table 8.

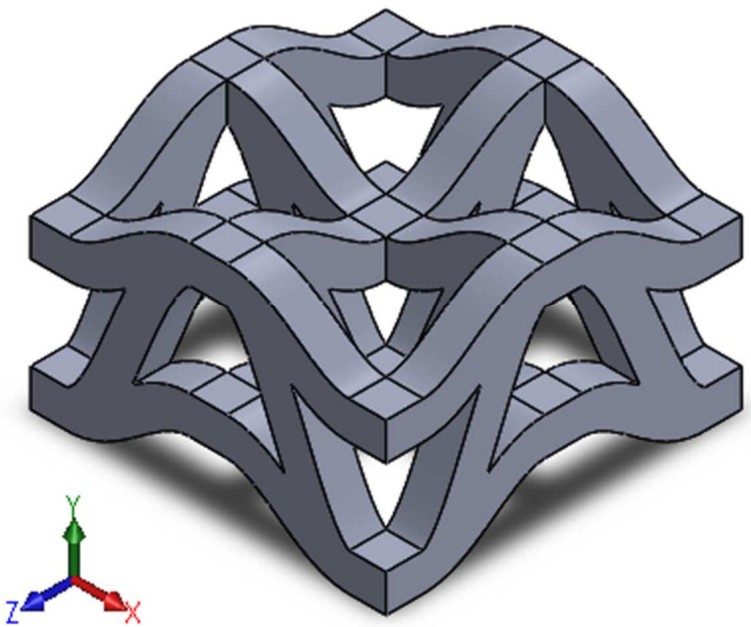

**Figure 7.** CAD for optimized model.

**Table 8.** Geometric parameters of the optimum model.

| Geometry | L (mm) | W (mm) | H (mm) | Thickness (mm) |
|:---:|:---:|:---:|:---:|:---:|
| DUH | 10 | 17 | 3 | 2 |

To verify the prediction, the optimized model receives the same treatment as the other models, where it experiences quasistatic compression with a maximum displacement of 70% of the optimized cell's L ($70\% \times 10$ mm). The simulation will also be performed three times according to the noise level. The result is then compared with the average of the existing models and the prediction of the optimized model, summarized in Table 9.

**Table 9.** Verification of optimized model.

| Parameters | Existing (Average) | Prediction | Verification | Difference Verif-Existing | Difference Verif-Prediction |
|:---|:---:|:---:|:---:|:---:|:---:|
| Mean SEA (kJ/kg) | 19.29 | 40.62 | 42.19 | 22.91 | 1.57 |
| S/N Ratio (dB) | 24.58 | 33.76 | 32.45 | 7.87 | −1.31 |

The geometry shape of the cell is variable, with the highest contributor with 48.37% contribution, the cross-section's thickness as the second highest with 35.66% contribution, the length of the cell as the third highest with 5.39% contribution, the width of the cell as the fourth highest with 4.77% contribution, and the bending height is the lowest contributor with 4.21% contribution.

According to Pratomo [31], in the design for six sigma (DFSS), the difference or gain in the S/N ratio between the optimized and existing model must exceed 1 to be considered an excellent improvement. That means this optimization is considered a remarkable improvement (S/N ratio's gain equals 7.87).

## 3. Battery System Materials, Methods, and Results

The battery system simulation is performed to know the best configuration and the optimized cell structure's effectiveness in keeping the battery safe from ground impact.

### 3.1. Battery Validation

3.1.1. Battery Validation Modeling

Pouch battery is used as the specimen to be protected. The numerical model for the pouch battery is based on the punch indentation simulation from reference Sahraei et al. [27].

The general configuration of the recreation simulation consists of (from top to bottom) the indenter, battery cell, and floor, where the distance between each part is set to be as close as possible without pre-penetration. A rigid shell sphere with a radius of 6.35 mm is used as the indenter, and a rigid shell plane is used as the floor, both with element size of 1 mm and thickness of 1 mm. Figure 8 illustrates the general configuration of this model.

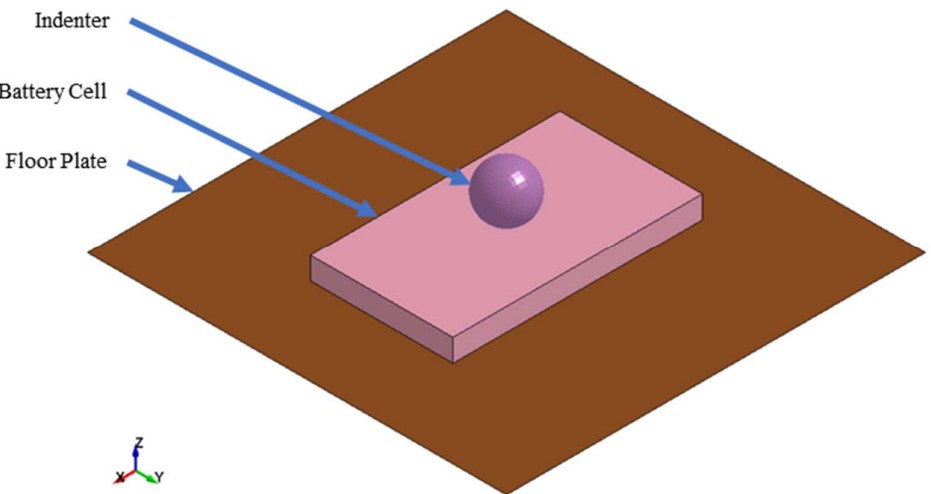

**Figure 8.** The general configuration of battery simulation; indenter as purple sphere, battery as pink box, floor as brown plane.

Here, the whole cell has the dimension of 59.5 × 34 × 5.35 mm and was modeled using fully integrated solid elements with element size of 1 mm in length and width direction and 0.5944 mm in the thickness direction (thickness of the battery divided by 10) with crushable foam material. Based on the reference Sahraei et al. [27], the behavior in the thickness direction is mostly a function of the active material and binder properties rather than the aluminum/copper foils. Therefore, the battery model can be modeled as uniform material without modeling the foils. Figure 9 illustrates the battery's cell geometry.

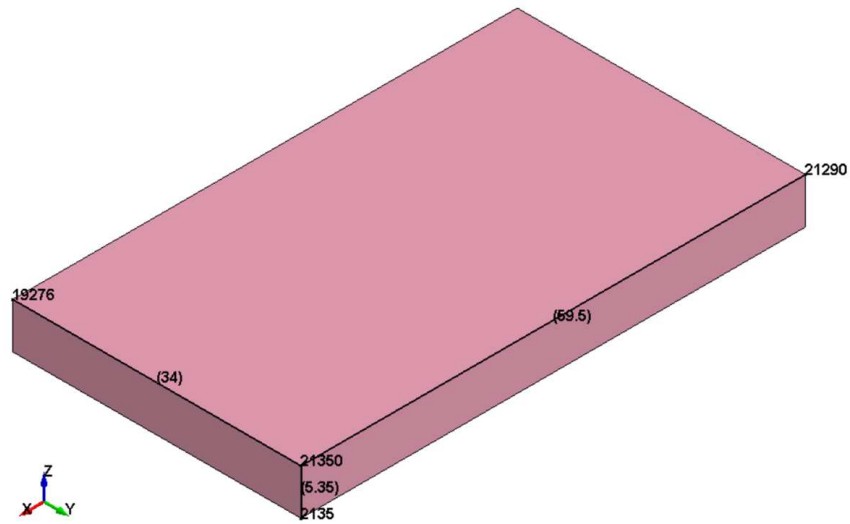

**Figure 9.** Battery cell's geometry.

The simulation is performed in a quasistatic process, just like the battery simulation. The velocity will increase according to Table 2, with the final velocity of 1 m/s in the direction of $Z - (\mathbf{DOF} = 3)$. The movement of every node of floor elements is constrained in every direction, and the movement of every node of indenter elements is constrained in every direction but the Z direction.

For the material properties, the modulus of elasticity for tensile loading is the maximum slope of the stress–volumetric strain curve ($E = 500$ MPa). The tensile cut-off value is taken from the average tensile strength of the battery cell, which is 55.58 MPa. The Poisson's ratio was assumed to be 0.01 due to the porosity of the battery's cell. The density of the foam was assumed to be $1.7555 \times 10^{-6}$ Kg/mm$^3$, which is calculated from the battery's weight (19 g) and volume (10, 823.05 mm$^3$). Table 10 and Figure 10 will display the material property of the battery's foam.

**Table 10.** Material property of battery's foam [27].

| Variable | Value | Unit |
|---|---|---|
| Density ($\rho$) | $1.76 \times 10^{-6}$ | Kg/mm$^3$ |
| Modulus of Elasticity ($E$) | 500 | MPa |
| Tensile Cut-off | 55.58 | MPa |
| Poisson's Ratio ($v$) | 0.01 | |
| DAMP Factor | 0.5 | |

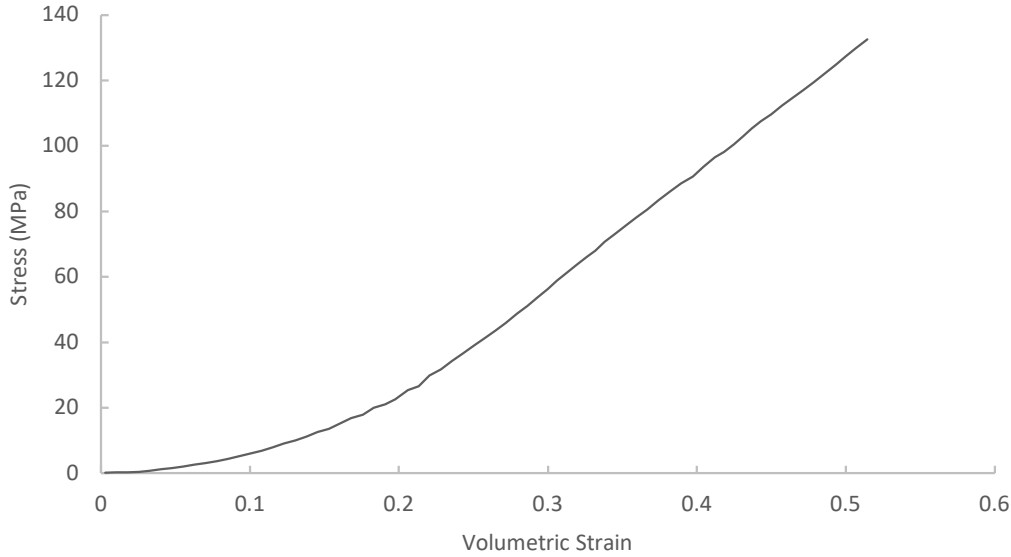

**Figure 10.** Experimental stress–volumetric strain curve through the thickness unconfined compression [27].

The damping factor is set to be 0.5 (maximum recommended value) due to its tendency to make the simulation more stable and the result of the convergence test, as shown in Figure 11.

Automatic single surface was applied for the contact between elements from the same part (every part). Automatic surface to surface was applied for the contact between different parts (battery–indenter and battery–floor). Interior was applied for the contact between elements inside the battery.

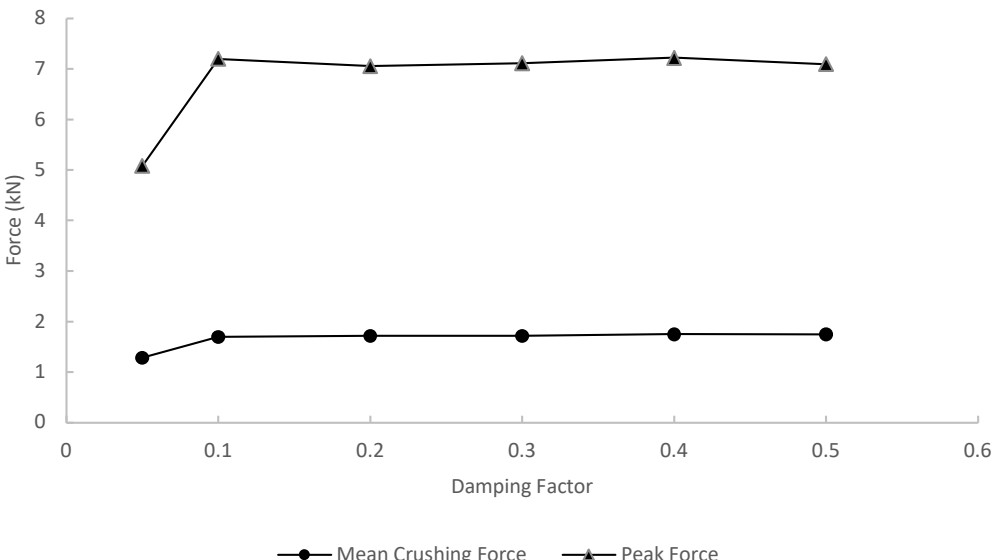

**Figure 11.** Convergence test of damping factor on battery model.

### 3.1.2. Battery Validation Result

Based on reference Sahraei et al. [27], it is found that the short circuit will occur on this pouch battery when it is compressed at about 2.9 mm. The goal of the numerical model is to recreate a similar force–displacement curve as the experimental result to the point of failure (0 to 2.9 mm displacement). Figure 12 shows the force curve of the numerical result compared to the experimental result.

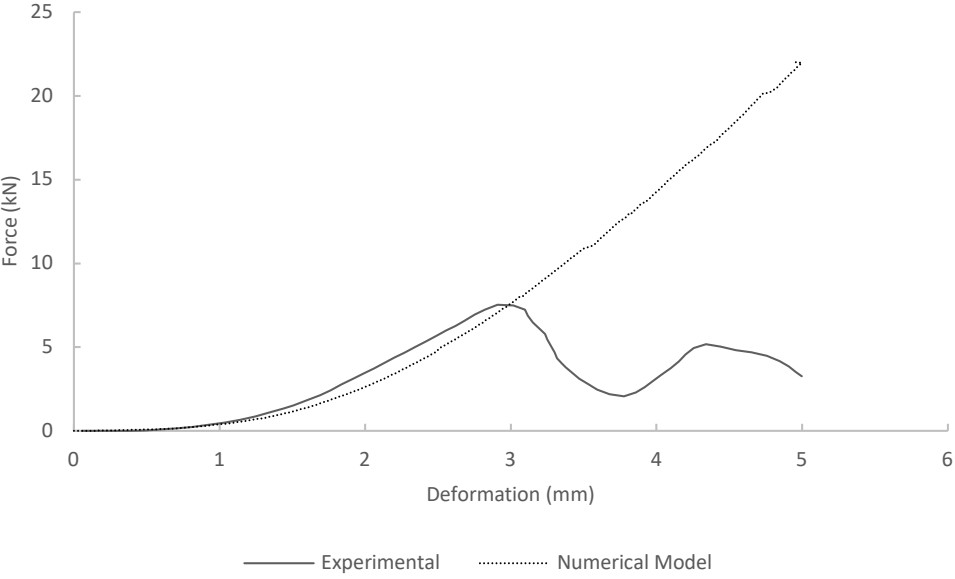

**Figure 12.** Force curve of the pouched cell battery.

With this result, the model is assumed to be accurate enough for this research.

### 3.2. System Modeling

The general configuration of the battery system consists of (from top to bottom) the floor, battery, upper plate, cell structure, lower plate, and impactor. Figure 13 illustrates the general configuration of this model.

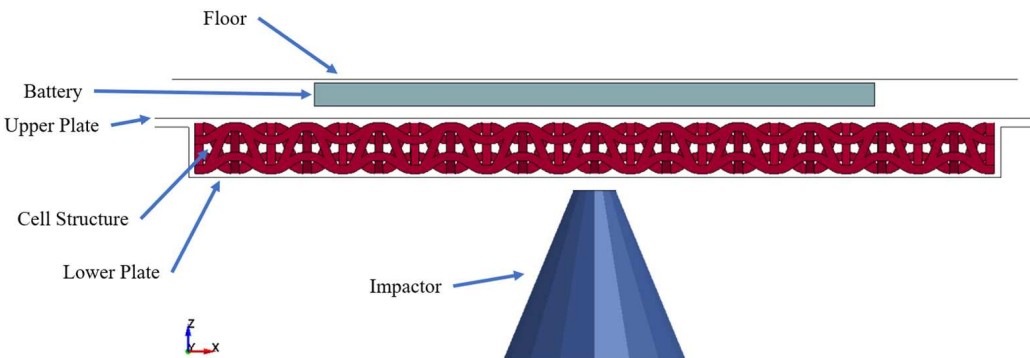

**Figure 13.** General configuration of the battery system.

The battery is developed from the recreation model from Section 3.1 which is also developed from reference Sahraei et al. [27]. The only change from the recreation model is the length and width of the battery, which became 132.5 mm, where this size is chosen for all configurations. The model still has the same thickness as the recreation model, 5.35 mm. It is assumed that the failure point of this battery is the same as the experimental result in reference Sahraei et al. [27], deformation of 2.9 mm. If any point of the battery experiences deformation of more than 2.9 mm, the battery and its protector system are considered as failed.

The floor panel acts as the base of the vehicle's structure, where no object below it is supposed to breach into the vehicle's main compartment. The floor panel is made of Al2024-T351 with a thickness of 2 mm. The floor panel is modeled with fully integrated shell element and piecewise linear plasticity material. Table 11 and Figure 14 will summarize the material property of Al2024-T351.

**Table 11.** Al2024-T351 material properties [24].

| Variable | Value | Unit |
|---|---|---|
| Density ($\rho$) | $2.78 \times 10^{-6}$ | Kg/mm$^3$ |
| Modulus of Elasticity ($E$) | 73.1 | GPa |
| Yield Strength ($\sigma_{ys}$) | 0.324 | GPa |
| Poisson's Ratio ($v$) | 0.33 | |
| Failure Strain ($\varepsilon_f$) | 0.2 | |

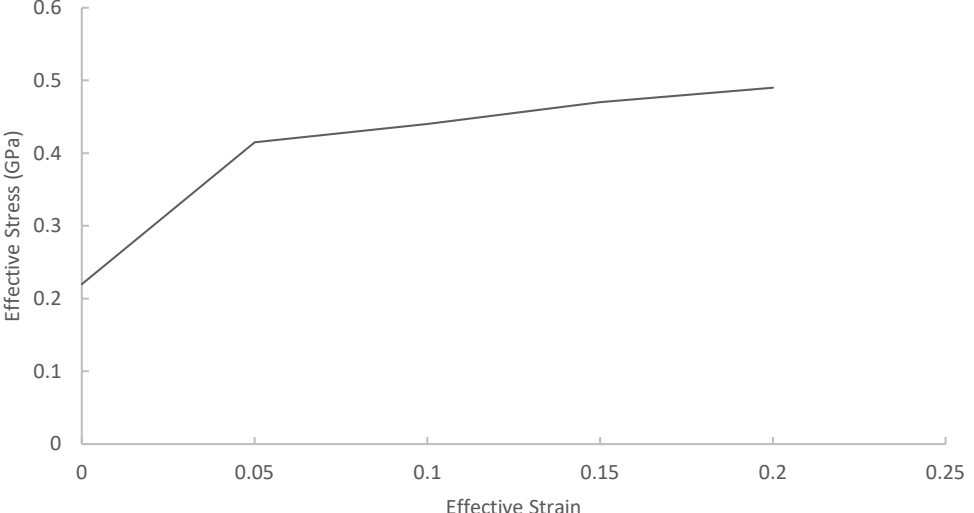

**Figure 14.** Al2024-T351 effective stress–strain curve.

The battery protector consists of a combination of upper plate, cell structure, and lower plate that act as a sandwich structure with cell structure as the core. Both the upper plate and lower plate are made of Al2024-T351 and modeled with fully integrated shell element and piecewise linear plasticity material. The dimension of both the upper and lower plate is summarized in Figure 15.

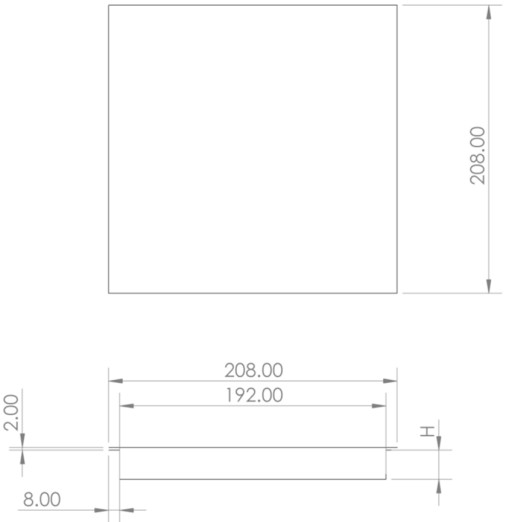

**Figure 15.** Upper and lower plate dimensions [25].

In this optimization, there are three types of cell structure configurations. Table 12 and Figure 16 show the dimension of each configuration.

**Table 12.** Cell structure configurations.

| Config. | Cell Resize | Cell Arrangement | Total Dimension | Structural Volume |
|---|---|---|---|---|
| First Config. | 100% length | $11 \times 11 \times 1$ cells | $189 \times 189 \times 12$ mm | $96,484.51$ mm$^3$ |
| Second Config. | 200% length | $5 \times 5 \times 1$ cells | $174 \times 174 \times 24$ mm | $168,488.23$ mm$^3$ |
| Third Config. | 300% length | $3 \times 3 \times 1$ cells | $159 \times 159 \times 36$ mm | $218,307.51$ mm$^3$ |



**Figure 16.** CAD of the first (**left**), second (**middle**), and third (**right**) configuration.

All cell configurations are modeled the same as in Section 2.2.3. The element size of each configuration is equal to the thickness of the individual cell divided by 3. This mesh density is the same as the validation model (see Section 2.1), yet different from the optimization model (see Section 2.3). Validation ensures that this reduction in mesh density compared to the optimization model will not significantly change the numerical result. The validation result is shown in Table 13 and Figure 17. There is only a notable difference in the plastic region. However, due to the limitation of time and computing power, this result is considered accurate enough for battery system simulation.

**Table 13.** Performance and property of the simplified and original optimum model.

| Parameter | Noise | | | Simplified |
|---|---|---|---|---|
| | **+1** | **0** | **−1** | **Simplified** |
| Volume (mm³) | | 1187.43 | | |
| Mass (g) | 5.52 | 5.26 | 5.00 | 5.26 |
| EA Total (J) | 214 | 225 | 225 | 190 |
| SEA (kJ/Kg) | 38.75 | 42.78 | 45.03 | 36.13 |

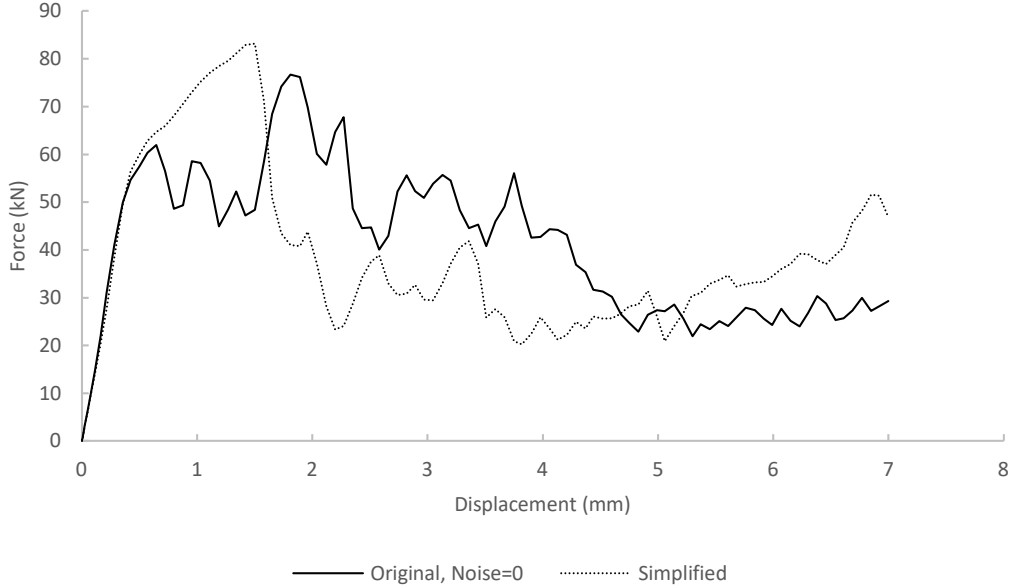

**Figure 17.** Force curve of the simplified and original optimized model.

The cell structure is made from the same material as the single-cell model, Ti-6Al-4V (see Table 7). The impactor model is modeled as a cone with a flat tip and a conical angle of 45°, with a fully integrated solid element with rigid material.

The impactor is designed to weigh 0.77 Kg and with an initial velocity of 42 m/s in the direction of $Z + (\mathbf{DOF} = 3)$. The impactor's initial kinetic energy is equal to the whole system's total energy throughout the simulation, calculated by Equation (2).

$$E_{total} = E_{k\ initial} = \frac{1}{2}mv^2 = \frac{1}{2} \times 0.77 \text{ Kg} \times \left(42\frac{\text{m}}{\text{s}}\right)^2 = 679.14 \text{ J} \tag{2}$$

*3.3. Result and Discussion*

Figure 18 shows the numerical results of all configurations.

The best configuration of the sandwich-based 3D auxetic structure for battery protection is chosen according to several requirements. First, it must successfully protect the battery from the ground impact loading, which means that it must not fail during the simulation. Second, it must be as light and small as possible because the battery protector is preferable to be as light as possible because lighter vehicles consume less energy under the same condition [10].

The first configuration is the only one that fails to protect the battery. The battery is deformed more than the deformation threshold for battery failure and even becomes punctured in some parts. This means that both the second and third configurations pass the first requirement. The dimension and mass of the structure are compared to choose between the second and third configuration.

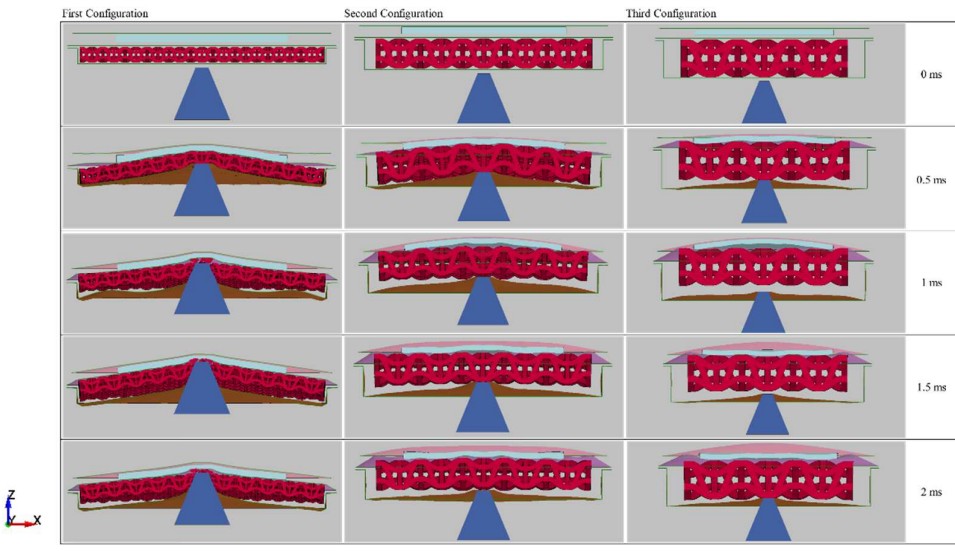

**Figure 18.** Numerical result of the first (**left**), second (**middle**), and third (**right**) configuration.

As we can see from Table 14, the second configuration is much shorter and lighter (33.33% and 22.82% less, respectively) than the third configuration. The second configuration's cell structure's absorbed energy is even larger by 1.03% than the third configuration's cell structure, resulting in a 30.90% larger SEA. Based on this value, the best cell configuration to protect this battery is the **second configuration**. Figure 19 shows the illustration of the second configuration.

**Table 14.** Comparison between configurations' numerical results.

| Configuration | Volume (mm$^3$) | Mass (Kg) | EA (J) | SEA (kJ/Kg) | Max Battery Deform (mm) |
|---|---|---|---|---|---|
| First Configuration | 96,484.51 | 0.43 | 531.00 | 1.24 | FAIL |
| Second Configuration | 168,488.23 | 0.75 | 591.00 | 0.79 | 1.92 |
| Third Configuration | 218,307.51 | 0.97 | 585.00 | 0.61 | 1.36 |

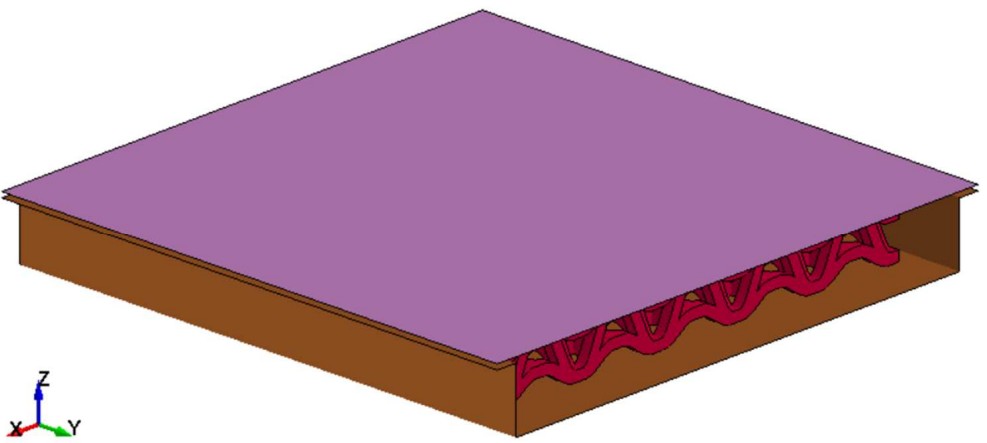

**Figure 19.** Illustration of the second configuration.

Another interesting observation to make in Table 14 is that the configuration consisting of smaller single-cell structures tends to have a larger SEA; yet, the battery that it is protecting also experiences more deformation. Although the volume and mass of the first configuration are 55.80% smaller than the third configuration, there is only a 9.23% difference between the energy absorbed by the first configuration and the third configuration.

This difference resulted in a 105.38% increase in SEA for the first configuration from the third configuration.

Although the SEA is inversely proportional to the size of the single-cell structure, the battery's deformation is directly proportional to the size of the single-cell structure. The writer suspects that this is due to the higher stiffness of the configuration's structure with a larger-size single-cell structure because of the higher thickness of the overall sandwich structure. This is the same reason why sandwich structures with a larger thickness have higher flexural stiffness and are very good at withstanding three-point bending, according to Campbell [12].

## 4. Discussion and Conclusions

The contribution of each 3D auxetic structure dimension parameter for SEA is calculated with the analysis of variance (ANOVA) (see Section 2). There are five dimension parameters that are analyzed, where it is found that the geometrical shape of the cell is the highest contributor with 48.37% contribution, the cross-section's thickness as the second highest with 35.66% contribution, the length of the cell as the third highest with 5.39% contribution, the width of the cell as the fourth highest with 4.77% contribution, and the bending height is the lowest contributor with 4.21% contribution, leaving just 1.60% contribution for error. Although the contribution varies greatly between each parameter, all five parameters are considered statistically significant.

The Taguchi orthogonal array calculates the most optimum dimension of the single-cell 3D auxetic structure for the highest SEA output (see Section 3). It is found that a double-U hierarchal/auxetic geometric shape, with 10 mm of length, 17 mm of width, and 3 mm of bending height, with 2 mm of cross-section's thickness, is the optimum geometry dimension for the highest SEA output. When proceeding through compression loading in a numerical simulation, the SEA output of the optimized model is between 38.75 kJ/Kg and 45.03 kJ/Kg (depending on the noise level) with a mean of 42.19 kJ/Kg. This is an increase of 4.50 kJ/Kg from the highest of the 16 existing models.

Based on the results in Section 4, the best cell configuration is the second configuration, where the optimized cell is enlarged to 200% in length (single cell's dimension $38 \times 38 \times 24$ mm), arranged in $5 \times 5 \times 1$ cells resulting in total dimension $174 \times 174 \times 24$ mm. This dimension resulted in a volume of $168,488.23$ mm$^3$ with a mass of 0.75 Kg. With this configuration, the maximum deformation of the battery is 1.92 mm, which is much lower than the deformation threshold for battery failure (2.9 mm).

As mentioned in Section, the higher stiffness is more favorable than the energy absorption capability of the overall structure for protecting the pouch battery from ground impact loading. These finds may be irrelevant if the cell structure and the pouch battery have other failure criteria, so further research is needed to validate this finding.

**Author Contributions:** Conceptualization, S.P.S.; methodology, S.P.S., L.G. and D.W.; numerical computation, M.A.S.B.; validation, M.A.S.B., S.P.S., L.G. and D.W.; formal analysis, M.A.S.B., S.P.S., L.G. and D.W.; investigation, M.A.S.B.; resources, S.P.S., L.G. and D.W.; data curation, M.A.S.B.; writing—original draft preparation, M.A.S.B.; writing—review and editing, S.P.S., L.G. and D.W.; visualization, M.A.S.B.; supervision, S.P.S., L.G. and D.W.; project administration, S.P.S., L.G. and D.W.; funding acquisition, S.P.S. All authors have read and agreed to the published version of the manuscript.

**Funding:** This research was supported by the ITB Research Program (2022).

**Data Availability Statement:** The data presented in this study are available on request from the corresponding author.

**Acknowledgments:** This research is fully funded by ITB Research Program managed by the Center for Research and Community Service (2022).

**Conflicts of Interest:** The authors declare no conflict of interest.

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
