# Peer review of "Design and Optimization of Lightweight Lithium-Ion Battery Protector with 3D Auxetic Meta Structures"

_wevj, doi:10.3390/wevj13070118_

Round 1

Reviewer 1 Report

This study presents a design and optimization of the sandwich structure to protect the pouch battery system from ground impact load. The sandwich structure design is based on auxetic structure. Detailed design, modeling, and simulation are provided. Generally, the topic presented in interesting. The work can be accepted for publication after addressing the following issues.

1, The language should be improved. Besides there are some writing errors in the manuscript, for example, "Each cell has the dimension of".

2, In Fig.2, why is only a sub-figure (a) presented? And what are the values for h1 and h2?

3, Regarding the analysis of the result difference shown in Fig.4, it is not acceptable. Could you please give more reasonable explanation on this?

4, What is the constraint condition for the cell optimization?

5, The error of the force curve obtained by model is large, why? How to reduce the influence by the large error?

6, With the design of the sandwich structure, how much will the energy density of the battery pack be influenced?

Reviewer 2 Report

The study in this manuscript presented a design of Li-ion battery protector with 3D auxetic meta structures. The authors did not connect the background of the topic in terms of the importance of the study and the previously conducted work with the intended work of this study.

Another important point in this study is the usage of the word “optimization” in the title. The study misses the whole structure of the optimization such as objective function, control variables, constraints, solution method, and validations. It is highly recommended to avoid such usage.

The reviewer would also suggest the following changes to be addressed and applied to maintain a comprehensive study:

1.      Line 28: the keywords should be arranged alphabetically

2.      Line 70: thesis?

3.      Lines 75-81: this section requires much more elaboration to cover the literature review and main discussions of previous studies conducted.

4.      Line 98: “each cell has the dimensions of” …. It seems that the rest of the information is missing!

5.      Please highlight your contributions with respect to Figure 2 and Table 1. Are they both used in total as in reference [18]? If yes, what are your contributions here?

6.      The previous point also applied to Figure 3 and Table 3, Table 10 and Figure 10, Table 11, Figure 15, Figure 21. For all of these cases, please explain what exactly is cited and for what purpose.

7.      Figure 5: the presented reference models and the writer’s model do not seem to be comparable in this context. Would you please justify how they are comparable? Especially with the used heights (10, 20, 30, and 40 mm) while the reference was shown at (10, 20, 40, and 60 mm). Also, what is the reference you are referring to here?

8.      Figure 6: it is better to use letter notations for the subfigures as (a, b, c ….)

9.      What are your thoughts with regards to Table 6? Any discussions?

10.  Section 4 has many cases of “Error! Reference source not found” please fix them.
